

# Identification of differentially expressed methylated genes in melanoma versus nevi using bioinformatics methods

Congcong He[1], Yujing Zhang[1], Hanghang Jiang[2], Xueli Niu[1], Ruiqun Qi[1] and Xinghua Gao[1]

[1] Department of Dermatology, The First Hospital of China Medical University and Key Laboratory of Immunodermatology, Ministry of Health and Ministry of Education, Shenyang, Liaoning, China
[2] Department of Dermatology, Shengjing Hospital of China Medical University, Shenyang, Liaoning, China

## ABSTRACT

**Background.** Melanoma is a highly invasive malignant skin tumor. While melanoma may share some similarities with that of melanocytic nevi, there also exist a number of distinct differences between these conditions. An analysis of these differences may provide a means to more effectively evaluate the etiology and pathogenesis of melanoma. In particular, differences in aberrant methylation expression may prove to represent a critical distinction.

**Methods.** Data from gene expression datasets (GSE3189 and GSE46517) and gene methylation datasets (GSE86355 and GSE120878) were downloaded from the GEO database. GEO2R was used to obtain differentially expressed genes (DEGs) and differentially methylation genes (DMGs). Function and pathway enrichment of selected genes were performed using the DAVID database. A protein-protein interaction (PPI) network was constructed by STRING while its visualization was achieved with use of cytoscape. Primary melanoma samples from TCGA were used to identify significant survival genes.

**Results.** There was a total of 199 genes in the hypermethylation-low expression group, while 136 genes in the hypomethylation-high expression group were identified. The former were enriched in the biological processes of transcription regulation, RNA metabolism and regulation of cell proliferation. The later were highly involved in cell cycle regulation. 13 genes were screened out after survival analysis and included: ISG20, DTL, TRPV2, PLOD3, KIF3C, DLGAP4, PI4K2A, WIPI1, SHANK2, SLC16A10, GSTA4O, LFML2A and TMEM47.

**Conclusion.** These findings reveal some of the methylated differentially expressed genes and pathways that exist between melonoma and melanocytic nevi. Moreover, we have identified some critical genes that may help to improve the diagnosis and treatment of melanoma.

Corresponding authors
Ruiqun Qi, xiaoqiliumin@163.com
Xinghua Gao, gaobarry@hotmail.com

# INTRODUCTION

Melanoma is a highly invasive malignant skin tumor derived from melanocytes whose incidence has been showing yearly increases. It can be induced by many factors, of which
ultraviolent light is considered to be the most common environmental factor, along with a strong heredity component (*Rastrelli et al., 2014*). In contrast to melanoma, nevus is a pigmented lesion formed by transformed melanocytes and is a common benign tumor in the skin. While most melanomas are primary tumors, approximately 33% of ALL PRIMARY MELANOMAS arise from nevi (*Damsky & Bosenberg, 2017*). In this way, although melanoma and melanocytic nevi may share some common biological characteristics, of perhaps greater significance, are their differences, which represents a rich area of investigation.

Most past research on melanoma has mainly focused on genomic variations, among which is the mutated BRAF gene, which is highly related to melanoma (*Bruno et al., 2017*). This work has resulted in substantial contributions for the diagnosis and treatment of melanoma. Current efforts and attention have been directed to the study of cancer epigenetics, a promising research field in discovering tumor developing mechanisms and biomarkers. Epigenetics refers to the effects of genetic alterations other than DNA variations on gene expression and function, including DNA methylation, non-coding RNAs and histone remodeling (*Baylin & Jones, 2011*). CpG dinucleotides, which tend to gather into clusters called CpG islands (*Straussman et al., 2009*), represent one such example. Methylation of these CpG islands is associated with the silencing of many genes, with DNA methylation affecting gene expressions that play important roles in the development of melanoma, such as LINE-1, TERT, MGMT, KIT, TNF and MITF (*Fu et al., 2017*). However, the mechanism and pathways involved in DNA methylation are still not well understood.

Recently, microarray has been increasingly used in the exploration of the genetics and epigenetics of melanoma, with the results of these assays providing new perspectives about the etiology, treatment and prognosis of melanoma. Many differential genes and key pathways have been unearthed through bioinformatics analysis. As for epigenetics, other than methylation, LncRNA and microRNA, represent the most significant topics of investigation. However, to date, in the bioinformatics analysis of melanoma, no examples exist regarding combinations of aberrant methylation and gene expression in melanoma versus nevi. We have explored four data sets in the GEO database, two of which show differences in gene expression in melanoma and nevi samples (GSE3189, GSE46517), the other two show DNA methylation differences between them (GSE86355, GSE120899). In this report, we used bioinformatic tools to analyze related functions and pathways of differential genes, as well as their mutual interactions. Our goal was to identify novel melanoma-related genes and their corresponding DNA methylations, providing new insights into the occurrence and development of melanoma.

## MATERIALS & METHODS

### Microarry data and data processing

Four data sets from the Gene Expression Omnibus (GEO, https://www.ncbi.nlm.nih.gov/geo/) of The National Center for Biotechnology Information (NCBI) were selected. GSE3189 (*Talantov et al., 2005*) and GSE46517 (*Kabbarah et al., 2010*) (platform: GPL96

**Table 1 Information for melanoma GEO datasets.**

| GEO series | Platform | Samples no. | | (REFS.) |
|---|---|---|---|---|
| | | melanoma | Nevi | |
| GSE3189 | GPL 96 | 45 | 18 | *Talantov et al. (2005)* |
| GSE46517 | GPL 96 | 31 | 9 | *Kabbarah et al. (2010)* |
| GSE120878 | GPL 13534 | 89 | 73 | *Conway et al. (2019)* |
| GSE86355 | GPL 13534 | 33 | 14 | *Wouters et al. (2017)* |

Affymetrix Human Genome U133A Array) relate to gene expression, and the DNA methylation microarrays are GSE86355 (*Wouters et al., 2017*) and GSE120878 (*Conway et al., 2019*), respectively (platform: GPL13534 Illumina Infinium HumanMethylation450 BeadChip array) (Table 1). Totally, 45 melanoma and 18 nevi samples were included in GSE3189 while 31 melanoma and 9 nevi samples were included in GSE46517. An additional 33 melanoma and 14 nevi samples were enrolled in GSE86355 while 89 melanoma and 73 nevi were enrolled in GSE120878 (detailed information is contained in Table 1). GEO2R, an online analysis tool built in the GEO website, was used to analyze raw data in order to identify DEGs and DMGs. $P$ value < 0.05 and $|t|>2$ were set as cut-off values. Then, the online tool, Bioinformatics & Evolutionary Genomics, was used to calculate overlap genes. Nevi samples were set as the control group. Overlap of hypermethylated genes and down-regulated genes were considered as the hypermethylation-low expression group, while hypomethylated genes and up-regulated genes that intersected were considered as the hypomethylation-high expression group. The above results were illustrated within the Venn chart on the website.

## GO term and KEGG pathway analysis of DEGs and DMGs

The DAVID knowledgebase (https://david.ncifcrf.gov/), an online gene functional annotation tool, was used to analyze the function and pathway enrichment of obtained DEGs (*Sherman et al., 2007*). The Fisher exact test $P$-value was calculated and a $P$-value < 0.05 was regarded as being statistically significant.

## PPI network and hub gene analysis

For analyses of interactions among proteins of interest, the STRING platform, an online tool for the structural and functional analysis of protein interactions (*Szklarczyk et al., 2017*) was used, 0.4 was regarded as the cut-off criterion and the active interaction sources were Textmining, Experiments, Databases, Co-expression, Neighborhood, Gene Fusion and Co-occurrence. Then Cytoscape software 3.6.1 (https://cytoscape.org) and built-in app Molecular Complex Detection (MCODE) was used to identify core modules in these proteins.

## Survival analysis of differentially expressed genes

Data on gene expressions from melanoma patients were obtained from the TCGA data portal (https://tcga-data.nci.nih.gov/tcga/), information on RNA expression as well as survival data were downloaded. Only primary melanoma samples were enrolled. A

univariate Cox model was used to illustrate the overall survival of differentially expression genes, $p < 0.05$ was the cut off value.

## RESULTS

### DEGs and DMGs in melanoma

The four data sets were separately analyzed using GEO2R and data with a $P$-value $< 0.05$ and $|t|>2$ were then selected for further study. Among them, the hypermethylation-low expression group had 199 genes while the hypomethylation-high expression group had 136 genes (Fig. 1).

### GO functional enrichment and KEGG pathway analysis

The GO functional enrichment was analyzed with use of DAVID, and included biological process (BP), cellular component (CC) and molecular function (MF). The top ten of each functional group for both hypermethylation-low and hypomethylation-high expression genes are listed in Figs. 2 and 3. In the hypermethylation-low expression group, BP was mainly involved in the regulation of transcription and RNA metabolism, regulation of cell proliferation, gene expression, cell biosynthesis, nitrogen-containing compounds and macromolecular metabolism. CC enrichment indicated that the differentially expressed genes were mainly distributed in various parts of the plasma membrane including the basial plasma membrane, apical plasma membrane and the membrane raft, while cell junctions including anchoring junction, adherens junction and cell-substrate junctions were also involved. As for MF, transcription regulation was the main component, with other functions like protein dimerization, protein specific binding and sequence specific DNA binding also being involved (Fig. 2).

For hypomethylation-high expression genes, biological process enrichment indicated that the cell cycle phase was highly regulated, with immune responses, nuclear division, post-Golgi vesicle-mediated transport and organelle fission also being involved. These genes were mostly distributed in the cytoplasm including the spindle pole, endosome, cytosol, vacuole and organelle lumen. Molecular function showed the presence of binding with ATP, adenyl nucleotides and nucleosides (Fig. 3). The same tool was used for analyzing the KEGG pathway. But there were few significant pathways those genes enriched, the only one has significance is in the hypermethylation-low expression group , pathway of basal cell carcinoma involved.

### PPI network construction and module analysis

STRING database was used to construct a protein-protein interaction network, and with use of cytoscape, software key modules and hub genes were constructed. The 136 hypomethylation-high expression genes identified contained a notable module (Fig. 4), function of genes in this module is more likely to regulate cell cycle (Table 2), while genes in the hypermethylation-low expression group failed to show any significant modules (Fig. 5).

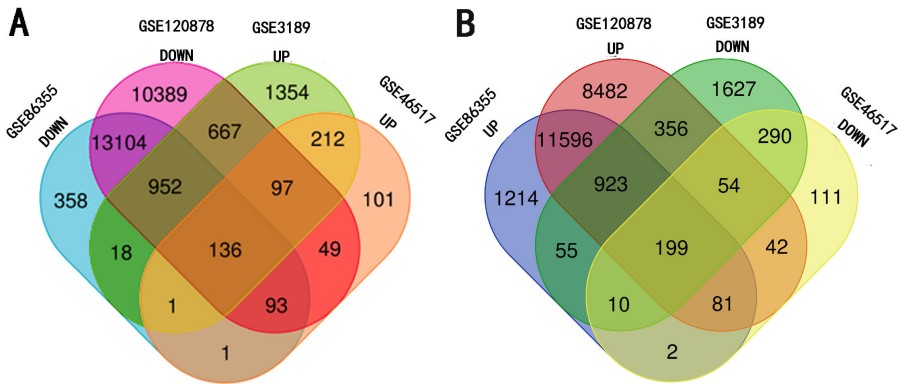

**Figure 1** Identification of methylated-differentially expressed genes in gene expression datasets (**GSE3189**, **GSE46517**) and methylation expression datasets (**GSE86355**, **GSE120899**). (A) Hypomethylation-high expression group. (B) Hypermethylation-low expression group.

## Survival analysis of differentially expressed genes

Information of primary melanoma samples from TCGA was used to screen out significant survival genes, then inserting with the differentially expressed genes above. Finally, 8 genes were selected in the hypomethylation-high expression group. They were ISG20, DTL, TRPV2, PLOD3, KIF3C, DLGAP4, PI4K2A and WIPI1. While 5 genes for hypermethylation-low expression group, including SHANK2, SLC16A10, GSTA4O, LFML2A and TMEM47. Interestingly, except DTL, significant survival genes belong to hypomethylation-high expression group were positively related to survival time. In contrast, In the hypermethylation-low expression group, they may had a negative effect to survival (Fig. 6).

## DISCUSSION

With continuous development of microarray and high-throughput sequencing technology, thousands of genes can be analyzed simultaneously, providing new insights for the diagnosis and treatment of melanoma. DNA methylation is a common epigenetic variation of melanoma and plays an important role in the development of this condition. In this study, we selected two mRNA expression data sets and two methylation data sets from the GEO database. After calculating the DEGs and DMGs, we combined hyper-methylation and low expression data and hypo-methylation and high expression in order to identify some candidate genes associated with melanoma pathogenesis and development. Then, GO and KEGG pathway enrichment analyses were performed based on these differentially expressed genes, and constructed protein-protein interactions were used to locate hub genes. Finally, we evaluated their effects on tumor survival.

In hypermethylation-low expression genes, transcription regulation was the most enriched function followed by nitrogen and macromolecular compound metabolism, regulation of cell metabolism and biosynthesis and protein binding. It seems clear that melanoma development is strongly associated with gene transcription, and a number
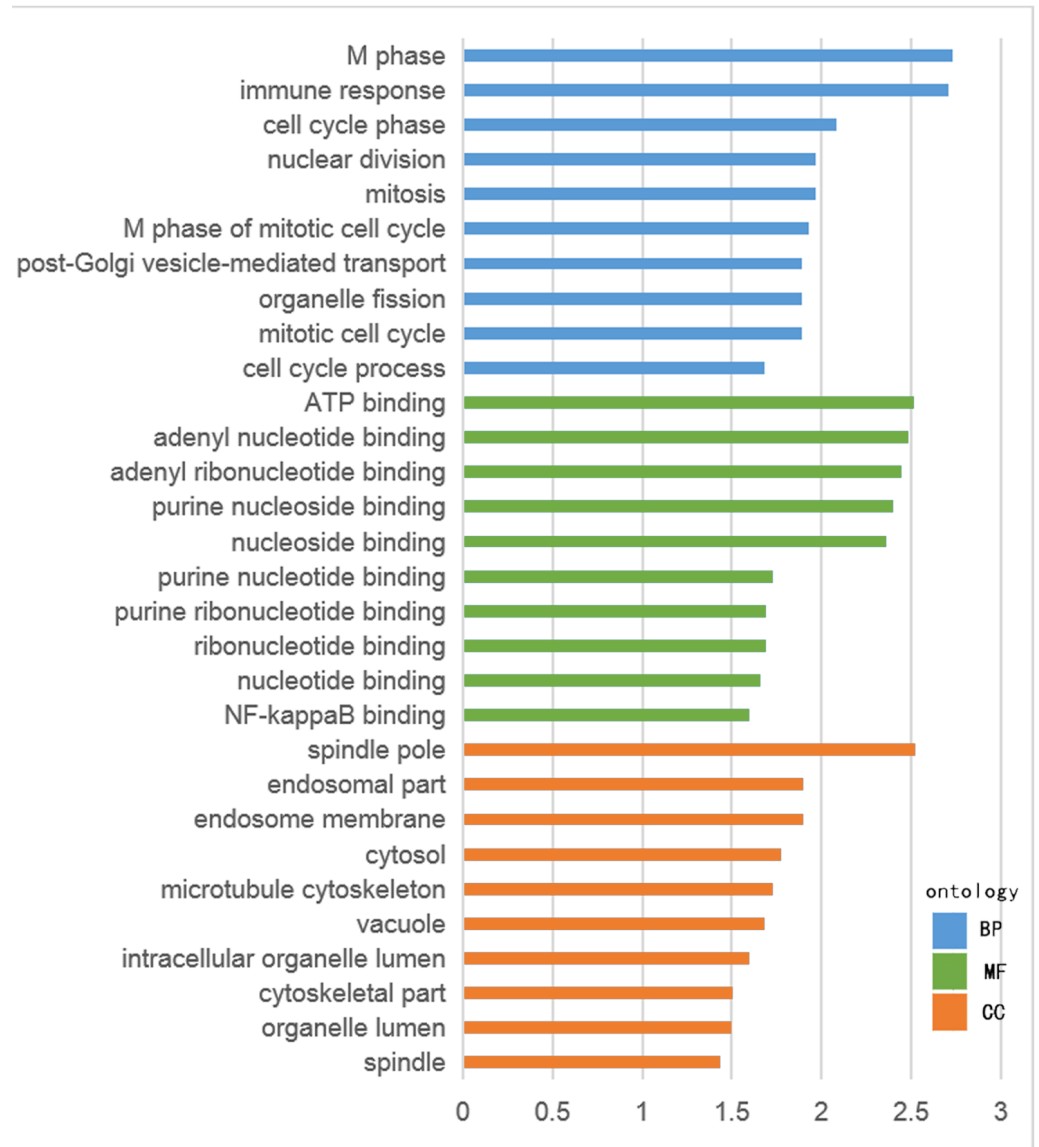

**Figure 2** GO analysis of hypomethylation-high expression genes.

of transcription factors are involved in its regulation. Among them, Microphthalmia-associated transcription factor (MITF) is the most common. Forced expression of MITF has been shown to be sufficient for induction of ectopic melanocytes in zebrafish (*Lister et al., 1999*) and the expression of at least some melanocyte differentiation genes in cultured mouse cells (*Hou, Arnheiter & Pavan, 2006*). In addition to MITF, other transcription factors such as *SOX10, YY1 and TFAP2A* (*Seberg, Van Otterloo & Cornell, 2017*) are involved with regulating the proliferation, differentiation and even invasion and metastasis of melanocytes. For the maintenance of melanoma, macromolecular substances such as sugar and protein are involved. For example, Heparan sulfate proteoglycans (HSPGs) participates in signal transduction by regulating ligands (*O'Connell & Weeraratna, 2011*).
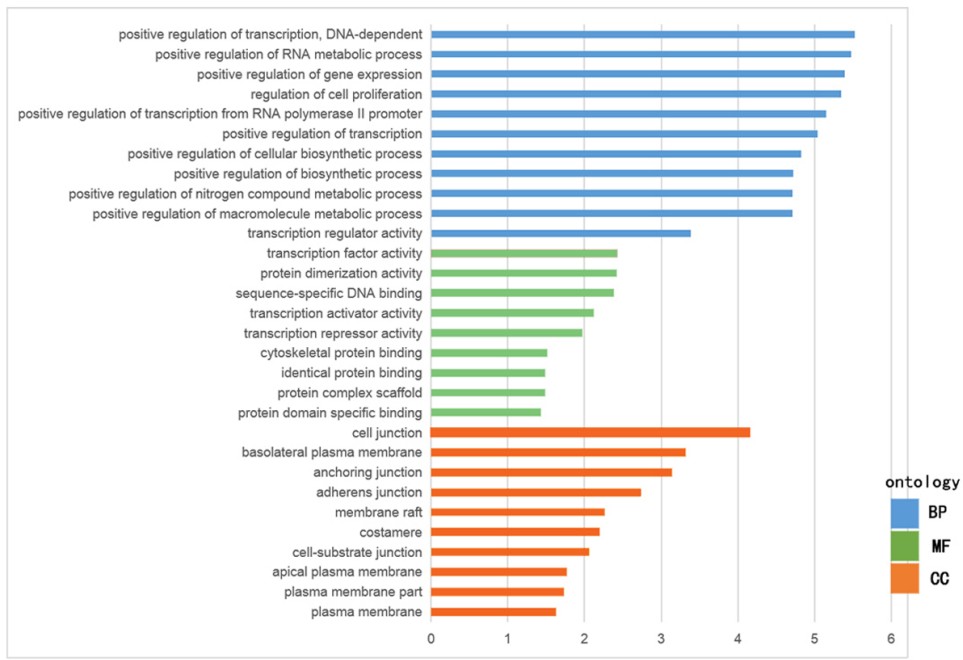

**Figure 3** GO analysis of hypermethylation-low expression genes.

Proto-oncogene-driven metabolic recombination is an adaptation of low energy in the tumor microenvironment, which serves to accelerate the proliferation of tumor cells (*Ratnikov et al., 2017*). The metabolism of macromolecular substances also provides energy for tumorigenesis. As most of these genes are concentrated on the plasma membrane in cells, component analysis, we suspect that many of these molecules are involved in cell signaling as membrane receptors.

In hypomethylation-high expression genes, cell cycle regulation, immune responses, and binding with ATP, represent the main processes exerted by these genes. As most of these genes are distributed intracellularly, and involve spindles, this is consistent with the metabolic processes of melanoma. Spindles are the organelles involved in mitosis, a process which requires ATP to provide energy. Regulation of the cell cycle is an important component of melanoma development, including such processes as cyclin-dependent kinase abnormalities and G1-S transition dysregulation (*Yao, Zuo & Wei, 2018*).

We performed a protein-protein interaction analysis on these genes and two significant modules were identified within the hypomethylation-high expression genes, of which, one module exhibited enrichment for several gene ontology terms which is mainly involved in the regulation of mitosis and cell cycle; and, our current results indicate that regulation of the cell cycle plays an important role during the development of melanomas.

The role of hypermethylation and demethylation genes in tumors is being further explored, but their function in the progressive ,metastasis process or protection effect is still not so clear. Studies have found that hypermethylation may be shared in primary disease and metastasis (*Micevic, Theodosakis & Bosenberg, 2017*), while loss of DNA

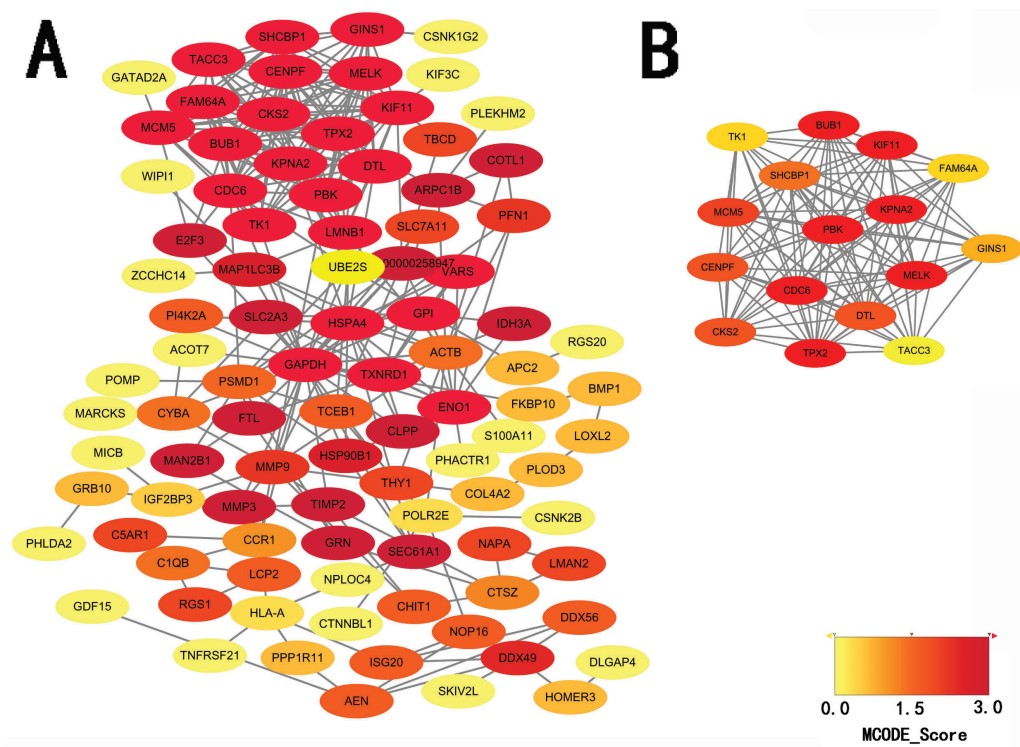

**Figure 4** **PPI network and top one module of hypomethylation-high expression genes.** (A) PPI network; (B) module 1.

**Table 2** **Top three GO terms and KEGG pathway involved in module 1 of hypomethylation-high expression genes.**

| Category | Term | gene | count | P Value |
|---|---|---|---|---|
| GO | BP | M phase | 8 | 5.324E−10 |
| | | cell cycle phase | 8 | 2.650E−09 |
| | | cell cycle process | 8 | 2.299E−08 |
| CC | | spindle | 5 | 2.475E−07 |
| | | spindle pole | 4 | 3.421E−07 |
| | | microtubule cytoskeleton | 5 | 4.713E−05 |
| | MF ATP binding | | 6 | 1.607E−03 |
| | adenyl ribonucleotide binding | | 6 | 1.710E−03 |
| | adenyl nucleotide binding | | 6 | 2.169E−03 |
| KEGG | hsa04110:Cell cycle | | 3 | 5.996E−04 |

methylation (hypomethylation) in cancer can lead to genomic instability, which can also occur in promoters of proto-oncogenes, leading to their activation and contribution to the progression of the malignancy (*Brait & Sidransky, 2011*; *Rodriguez-Paredes & Esteller, 2011*). In this study, we screened out genes significant to survival of melanoma patients, majority of them were verified to associated with tumors.
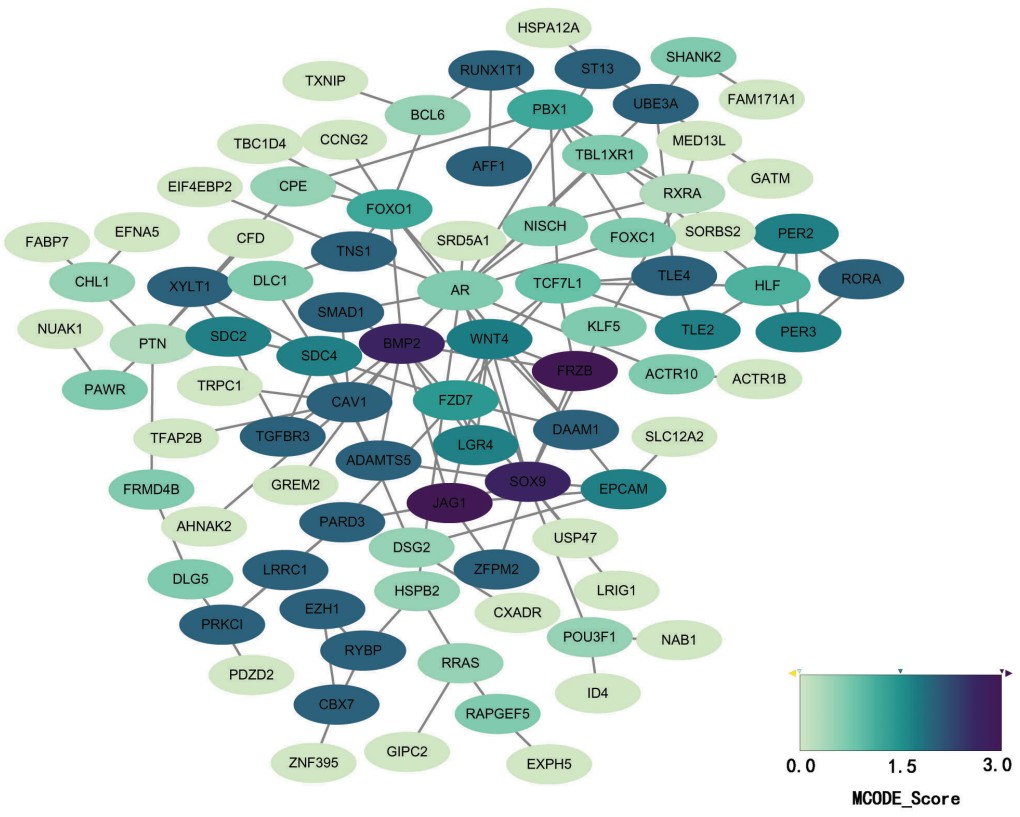

**Figure 5**   **PPI network of hypermethylation-low expression genes.**

ISG20 has interferon-related antiviral effects, while a study found that over-expression of ISG20 promotes metastasis and angiogenesis of liver cancer cells (*Gao et al., 2019*). DTL is involved with the regulation of cell cycles and playing a role in the regulation of DNA damage, so we suggest hat it regulate cycle of melanocytes so that to regulate its progression. TRPV2 is an ion channel, it's activation mediated melanoma cell death (*Zheng et al., 2019*), which may play a negative role in melanoma development. PLOD3 was found to be highly expressed in lung cancer and gliomas (*Tsai et al., 2018*), which is associated with poor prognosis of them. KIF3C is found to be highly expressed in breast cancer, KIF3C and SHANK2 are both related to axons transmitting neural signals (*Wang et al., 2015*), and poor prognosis of tumor, but its role in melanoma remains to be further studied. DLGAP4 is involved in the transmission of neuronal signals and has not been excavated for its role in tumors, but its epigenetic changes and dysfunction have been found to be related to early-onset cerebellar ataxia (*Minocherhomji et al., 2014*). PKR / PI4K2A lysosome network is associated with poor prognosis in breast cancer patients (*Pataer et al., 2019*). WIPI1 is an indicator of autophagosome formation, it also plays a distinct role in controlling the transcription of melanogenic enzymes and melanosome maturation. SLC16A10 can have function of a net efflux pathway for aromatic amino acids in the basosolateral epithelial cells. *Abel et al. (2010)* found hat GSTA4 is a novel susceptible gene for non melanoma skin

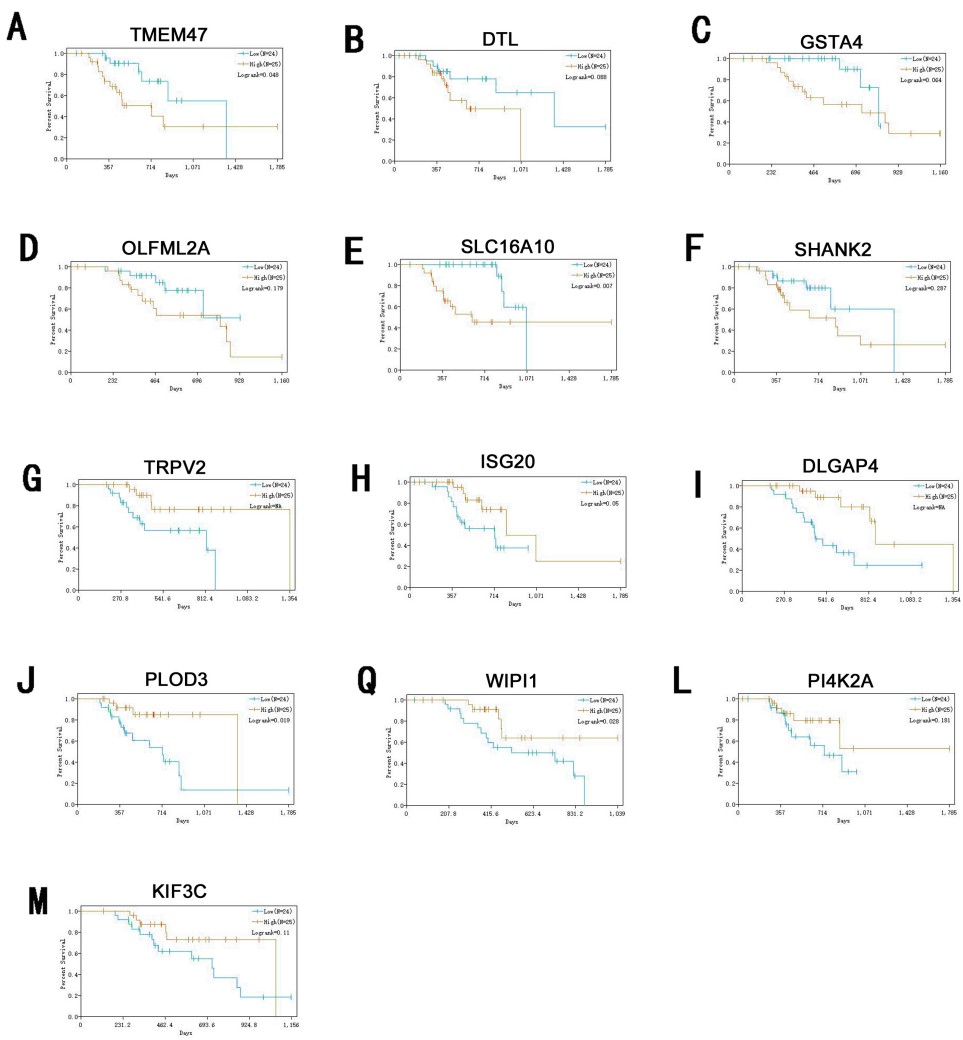

**Figure 6** **Survival analysis of genes significant to survival of melanoma patients.** (A) TMEM47. (B) DTL. (C) GSTA4. (D) OLFML2A. (E)SLC16A10. (F) SHANK2. (G) TRPV2. (H) ISG20. (I) DLGAP4. (J) PLOD3. (Q) WIPI1. (L) PI4K2A. (M) KIF3C.

cancer, although the relationship between GSTA4 and melanoma was not analyzed, but it still was suggested expression is associated with a poor prognosis for skin tumors. TMEM47 plays a role in regulating the localization of a subset of tight junction proteins, associated actomyosin structures, cell morphology, and participates in developmental transitions from adherens to tight junctions (*Dong & Simske, 2016*). But the role of OLFML2A has been seldom studied. While some of these genes have been previously studied, for many of them their specific role in melanoma remains unclear. Therefore, studies directed at investigating their roles in melanoma may provide new insights into the diagnosis and treatment of this condition.

## CONCLUSIONS

In conclusion, bioinformatics methods were used to identify aberrant methylation genes in melanoma relative to nevi. Specific survival genes that were revealed from this analysis include ISG20,DTL,TRPV2,PLOD3,KIF3C,DLGAP4,PI4K2A,WIPI1,SHANK2,SLC16A10, GSTA4O,LFML2A and TMEM47. The identification of such genes may play an important role in improving the diagnosis and treatment of melanoma. Further validation regarding their roles in melanoma will require additional molecular experiments.

### Funding

This study was supported by the National Key Basic Research Program of China [2013CB531604] (to Xinghua Gao), the Project for Construction of Major Discipline Platform in Universities of Liaoning Province [2015225012] (to Xinghua Gao), and the 111 Project [D18011] (to Xinghua Gao). The funders had no role in study design, data collection and analysis, decision to publish, or preparation of the manuscript.

### Grant Disclosures

The following grant information was disclosed by the authors:
The National Key Basic Research Program of China: 2013CB531604.
The Project for Construction of Major Discipline Platform in Universities of Liaoning Province: 2015225012.
The 111 Project: D18011.

### Competing Interests

The authors declare there are no competing interests.

### Author Contributions

- Congcong He conceived and designed the experiments, performed the experiments, analyzed the data, authored or reviewed drafts of the paper, and approved the final draft.
- Yujing Zhang analyzed the data, prepared figures and/or tables, authored or reviewed drafts of the paper, and approved the final draft.
- Hanghang Jiang and Xueli Niu analyzed the data, authored or reviewed drafts of the paper, and approved the final draft.
- Ruiqun Qi and Xinghua Gao conceived and designed the experiments, authored or reviewed drafts of the paper, and approved the final draft.

### Data Availability

    The analysed data is available in the Supplementary Files.

### Supplemental Information

Supplemental information for this article can be found online at http://dx.doi.org/10.7717/peerj.9273#supplemental-information.

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
