# Peer review of "Identification of differentially expressed methylated genes in melanoma versus nevi using bioinformatics methods"

_PeerJ, doi:10.7717/peerj.9273_

## Round 0.1 · original submission · Major Revisions

Please pay particular attention to the methodologic issues and concerns raised by both receivers, as these impact significantly on how the reader could be expected to interpret.correctly the data.

Reviewer 1 ·

Basic reporting

Please see general comments for the author

Experimental design

Please see general comments for the author

Validity of the findings

Please see general comments for the author

Additional comments

The authors present a bioinformatics study investigating the differentially expressed genes and differentially methylated genes in four datasets re-analysed from the NCBI GEO database. The authors used the NCBI GEO built in comparison tool (GEO2R), to identify significantly differentially expressed genes and significantly differentially methylated genes between benign nevi and malignant melanoma. The authors go on to define a “hypermethylation-low expression” group and a “hypomethylation-high expression” group which they subsequently subjected to gene ontology analysis, STRING analysis and hub gene analysis. Finally, the authors identified nine key hub proteins which they checked against TCGA and GTEx data using another online tool and found them to have a significant effect on survival.
There are several areas within this study that require major attention and/or clarification.

1. There is an entire dataset missing from the supplementary files. GSE3189, GSE86355 and GSE120878 are all present but the fourth dataset appears to be GSM46517 where it should be GSE46517 (GSE indicated a data series whereas GSM indicates just one sample). Either the wrong data set has been uploaded or the wrong dataset has been analysed from the start.

2. The Venn diagrams presented in figure 1 could not be reproduced with the data provided (disregarding the incorrect dataset provided outlined in 1st point) (appendix 1, incorrect dataset shaded in red). The entire datasets provided in the supplementary files were uploaded to the online tool the authors describe in their methods to recreate the Venn diagrams. Although no cut-off filter was used (as they outline in their methods), less overlap was achieved (highlighted with yellow circles), where equal or more overlap would be expected.

3. The STRING figures 4 and 5 appear different than would be expected from STRING or cytoscape. Figures 4 and 5 do not match the theme of either cytoscape or STRING and therefore the authors should specify how these networks were visually created. If they were produced using additional software, the authors should note this.

4. The authors report on non-significant findings without specifying their significance value. Lines 141:145 and 193:194; although the authors don’t specifically state significance, they should explicitly indicate that three of the five pathways they mention are not significant (table 2: pvalues=0.056, 0.085, 0.079)

Other areas of consideration include:
1. Line 86, authors state “corresponding DNA methylation microarrays are…”. This is misleading as it implies the methylation data has come from the same samples as the gene expression data, which I believe is not true.

2. No clarification of the parameters used for STRING analysis. The authors should clarify what parameters the string analysis was set to i.e. interaction score threshold and indeed, interaction sources used.

3. Line 159; “nine hub genes were screened out”. How were they screened out?

4. Table 3 appears in the manuscript after table 4.

5. Consider removing module 2 from figure 4(c). very weak module and the authors do not discuss.

6. Figure 1: the authors should make the labels in the same place for each Venn diagram to aid readability.

7. Figure 2 & 3: invert the figure. The lower p-values (i.e. higher log10 p-values) should be at the top of their respective groups, not the bottom. Furthermore, text size appears to be different between ontology groups, this should be rectified.

8. Figure 4: would aid in understandability if authors would highlight modules within the network. Furthermore, a scale is needed to indicated what the colours of each node mean (edges per node?).

In conclusion, I believe the most interesting aspect of this study is the identification of hub genes and their affect on survival in TCGA and GTEx data. Prior analyses, although some required to obtain the hub genes, are rather unremarkable and mundane. Furthermore, other studies have tried similar approaches, with no mention or comparison (PMID: 29302192, GEO:GSE31879).

Annotated reviews are not available for download in order to protect the identity of reviewers who chose to remain anonymous.

·

Basic reporting

The paper was well written with very few typos. The authors provided sufficient background knowledge and references for the context of the paper. The structure of the article is clear and the figures reflect the results generated from the experiments of the paper. All raw data was provided or was referenced to. Most of the results were relevant to the hypothesis of the paper although the methodology needs to be improved.

Experimental design

Uncovering the differences between nevi and primary melanoma is an important question in the melanoma field. As the transition from nevi to melanoma is not a very well understood process, this question is important and could have implications for the risk factors for developing melanoma. The authors attempted to integrate DNA methylation and gene expression data from published GEO datasets to address the differential methylation and gene expression between melanocytic nevi and primary melanoma to better understand the etiology and pathogenesis of primary melanoma.

The methods used in the paper were mostly correct. One of the main weaknesses in the methodology is not having paired samples for DNA methylation and gene expression. To circumvent this problem the authors analyzed 4 independent datasets separately to obtain the differentially expressed genes or methylated regions and then overlapped the gene sets.

For the target gene cut-offs, I would recommend using logFC as a cutoff instead of the moderated t-statistic, it is more biological meaningful. I would also suggest the authors perform further analysis of the differential methylation data. Using the moderated t-statistic or logFC as a cut-off is fine as a first step, but it doesn't reflect of the biological methylation status of the group since it is just a ratio and the absolute values of methylation aren't considered. Generally in DNA methylation, B values of less than 0.3 are considered hypo-methylation and B values of more than 0.7 are considered hypermethylation. For instance, if the FC between the groups is 3, it could signify a B value difference of 0.1 vs 0.3 or 0.3 vs 0.9. In the first scenario, I wouldn't consider it as a biological difference in the methylation status since 0.1 and 0.3 would both be considered as hypomethylation, whereas 0.3 vs 0.9 would give an actual biological difference in the methylation state. Due to this change, the authors will have to repeat the downstream analysis as the genes from the differential methylation analysis will change. It is also unclear how the authors took into account of the multiple probes from the methylation array that match to a single gene and if they considered CpG structures such as CpG Islands and Shores.

The authors use of cBioportal for validation is puzzling to me. As far as I know, the paired methylation data and gene expression data from cBioportal for melanoma are based from the TCGA which are tumor only samples and do not contain nevi. I don't think this is the correct dataset for validation as the authors want to look for correlation in nevi and primary. For survival analysis, the tool GEPIA uses TCGA data for survival analysis. Most of the melanoma samples in the TCGA are from metastatic patients, I would recommend the authors just use the primary samples for survival analysis, as it would be more relevant to their research question.

Validity of the findings

Since the methodology for differential methylation needs to be corrected and as well as the gene selection for differential gene expression using logFC, the current findings aren't interpretable.

Additional comments

I would recommend consulting a bioinformatician experienced in DNA methylation analysis to help analyze and interpret the differential methylation data. There are other specialized tools available for differential methylation analysis of the Illumina 450k platform. Although limma is a valid tool for log2 transformed B values

---

## Round 0.2 · Major Revisions

There remains some concern with the experimental design including the type of statistical analysis employed and whether methylation and gene expression datasets can be combined from different experiments. Please address these issues along with others in the reviewer's comments.

·

Basic reporting

no comment

Experimental design

I question the biological significance of the study if there were very few genes that passed the logFC cutoff. Although t-statistic can be used as a filter, as the authors suggested it has a tendency to choose genes with low expression. This would suggest that the differentially expressed genes between melanoma and nevi are not highly expressed which makes me doubt if these genes have a functional role in the disease.

Regarding the methylation analysis. There are several pipelines that analyze Illumina 450k data, here is are 2 review that summarizes a few of them.
https://dx.doi.org/10.1186%2Fs12859-018-2096-3
https://doi.org/10.1016/j.ymeth.2014.08.011

One of the issues still standing is the differential methylation analysis. Typically, CpG islands are more reliable than a single CpG probe and these CpG islands should be in the promoter regions of the gene of interest. I've attached a reference regarding this type of analysis. https://www.nature.com/articles/ng.865

Illumina provides a reference database for their probes in context to the gene. In the GSE86355 dataset this in under the column UCSC_RefGene_Group, GSE120878 is missing this column. I inspected the 9 hub genes in GSE86355 and CDC6, CENPF, GPI, VARS have significant down regulated probes located in the gene body or 3' UTR. See attached pdf.

I would suggest that the authors redo the methylation analysis by looking at differentially methylated regions by filtering for CpG probes in the 5'UTR and the TSS200, 1000 and 1500 regions. If the authors still prefer to analyze their data by probe, I would recommend to have at least more than 1 significant probe per gene as a very lax filter for hypo or hypermethylation.

As for the analysis of just the primary cases in the TCGA dataset. The authors can use this workflow to extract just the primary cases for SKCM from the GDC and analyze the survival differences. https://f1000research.com/articles/5-1542

Validity of the findings

no comment

---

## Round 0.3 · Major Revisions

While your revision has addressed several several of the key issues raised by the first review, some important ones remain outstanding. Addressing these in a satisfactorily manner will significantly improve your manuscript..

Reviewer 1 ·

Basic reporting

The background provided is sufficient to clarify the basis of the paper and the references are relevant and plentiful. Some clarification needed at points within the paper that lack good grammar (see comments for authors). Very few typos. The figures are relevant to the paper but the quality and resolution of figures 2 and 6 are very poor (I can’t read figure 6 plot titles to identify the genes).

Experimental design

The biggest issue with this paper is the inability to recreate the results from the data provided. The supplementary files for methylation and gene expression data offer different gene identifiers and so Venn diagrams obviously cannot be reproduced. It also warrants an explanation as to how the authors did this analysis themselves (with different identifiers). The authors should check the supplementary files (specifically the methylation data) and upload the exact data that was used to create the Venn diagram as until this point, it is unclear how the authors achieved the initial results, which subsequently puts the rest of the paper into question. I suggest the authors use gene symbols as the identifiers as most readers would prefer this format.

Nonetheless, as long as this clarification is addressed, the authors present a bioinformatics study which investigates the gene expression/methylation differences between naevi and primary melanoma ultimately identifying hypomethylated-high expression and hypermethylated-low expression genes of importance which they then go on to highlight that several have an effect on survival in melanoma patients. The analysis is straight forward and takes a sensible approach to answer the question at hand.

Validity of the findings

The authors describe their results with good detail and in a concise manner. The authors discuss their results and relate them to findings in the current literature to an acceptable degree. There are several issues with the proposed figures (outlined in comments for authors).

Additional comments

The area requiring attention from the authors are as follows:

Major:
The authors must provide the data used to create the venn diagrams so that the reader can reliably reproduce the results. To do this the identifiers must be consistent as at the moment, the expression uses gene symbol and the methylation data uses RefGene name.

Minor:
Line 50: The sentence “While most melanomas are primary tumours, approximately 33% develop from nevi” should be altered. I understand what the authors mean but all cancers arise from a primary tumour (even if that primary tumour arises from a naevus). Consider changing this to “approximately 33% of all primary melanomas arise from naevi”

Line 105: As per my previous recommendation, the authors have detailed the cutoff and evidence used in the STRING analysis. Whilst this is good, text mining, neighbourhood, gene fusion and co-occurrence is not ideal to use as evidence. I would suggest the authors try the analysis without these features and see how much it affects the results. If it makes little difference the authors should either state this or consider re-analysis. If the differences are major, the authors can leave the results as they are (the data is still reproducible as the authors have specified the parameters).

Line 138: The sentence beginning “But there are little significant pathways…” is grammatically incorrect and requires a re-write.

Line 148: The sentence beginning “Primary melanoma samples from TCGA was…” is grammatically incorrect and requires a re-write.

Line 192: I appreciate the authors removed the second module which had no significant enrichment but on line 192 where they state there are two significant modules, then only go on to talk about module one may confuse the reader and make them confused as to where the second is. I advise a clarification somewhere; i.e. “ two significant modules were identified within the hypomethylation-high expression genes, of which, one module exhibited enrichment for several gene ontology terms”.

Line 200: It is unclear what the authors mean in the sentence beginning; “in this study, we screened out genes….”.

Line 207: typo; glioms – glioma/gliomas

Figure 1: I advised in the previous review that the authors make the labels on the venn diagrams consistent between A and B as to aid in cross-reading between them. This has not been done. For clarification, the left most circle on both A and B plots should read the same GEO data set (i.e. GSE86355), so should the next circle, and the next, etc...

Figure 2 & 3: The authors have put figure 3 in decreasing -log10 pvalue, as per my initial recommendations but have not done this for figure 2, please could the authors do this. The axis titles have been cropped from both but need to be present for the viewer to understand the units. Furthermore, the image is of very low resolution and the text size is different between ontologies, this should be corrected and a higher quality image obtained.

Figure 4 & 5: The authors must provide a colour scale. The nodes of the network are colours but it is unclear what they are coloured by and so a scale or legend of some sorts is needed.

Figure 6: Almost unreadable. The authors must acquire higher quality images as it is almost impossible to read the graph titles and thus which graphs represent which genes. All text within this figure is extremely hard to read.

·

Basic reporting

The study is reported in a clear and understandable manner.

Experimental design

The authors justified their use of the t-statistic for the gene cutoff and performed survival analysis on only the primary cases in the TCGA melanoma dataset.

Validity of the findings

This analysis will be useful for future studies for identifying risk factors in genes associated with nevi and primary melanoma.

---

## Round 0.4 · accepted · Accept

Thank you for making the requested revisions, which I hope you will agree has strengthened your paper for more optimal reader accessibility.